# Haploinsufficiency of *SF3B2* causes craniofacial microsomia

Andrew T. Timberlake [1✉], Casey Griffin[2], Carrie L. Heike [3,4], Anne V. Hing[3,4], Michael L. Cunningham[3,4], David Chitayat[5,6], Mark R. Davis[7], Soghra J. Doust[8], Amelia F. Drake[9], Milagros M. Duenas-Roque [10], Jack Goldblatt[11], Jonas A. Gustafson [3], Paula Hurtado-Villa[12], Alexis Johns[13], Natalya Karp[14], Nigel G. Laing[15], Leanne Magee[16], University of Washington Center for Mendelian Genomics, Sureni V. Mullegama[17], Harry Pachajoa[18], Gloria L. Porras-Hurtado[19], Rhonda E. Schnur[17,20], Jennie Slee[11], Steven L. Singer[21], David A. Staffenberg[1], Andrew E. Timms[4], Cheryl A. Wise[7], Ignacio Zarante [22,23], Jean-Pierre Saint-Jeannet[2] & Daniela V. Luquetti[3,4✉]

Craniofacial microsomia (CFM) is the second most common congenital facial anomaly, yet its genetic etiology remains unknown. We perform whole-exome or genome sequencing of 146 kindreds with sporadic (n = 138) or familial (n = 8) CFM, identifying a highly significant burden of loss of function variants in *SF3B2* (P = 3.8 × 10$^{-10}$), a component of the U2 small nuclear ribonucleoprotein complex, in probands. We describe twenty individuals from seven kindreds harboring de novo or transmitted haploinsufficient variants in *SF3B2*. Probands display mandibular hypoplasia, microtia, facial and preauricular tags, epibulbar dermoids, lateral oral clefts in addition to skeletal and cardiac abnormalities. Targeted morpholino knockdown of *SF3B2* in *Xenopus* results in disruption of cranial neural crest precursor formation and subsequent craniofacial cartilage defects, supporting a link between spliceosome mutations and impaired neural crest development in congenital craniofacial disease. The results establish haploinsufficient variants in *SF3B2* as the most prevalent genetic cause of CFM, explaining ~3% of sporadic and ~25% of familial cases.

A full list of author affiliations appears at the end of the paper.

Craniofacial microsomia (CFM, MIM#164210), also termed hemifacial microsomia, oculo-auricular-vertebral spectrum (OAVS) or Goldenhar syndrome, comprises a variable phenotype, with the most common features including auricular malformations and underdevelopment of the mandible on one or both sides. Other frequently affected tissues include the middle ear ossicles, temporal bone, zygoma, and cranial nerves. Microtia in the absence of other anomalies is believed to represent the mildest form of CFM[1,2]. At the other end of the phenotypic spectrum, CFM can be associated with multiple anomalies including lateral oral clefts, epibulbar dermoids, and anomalies of the nervous, vertebral, renal, and cardiac systems. Multiplex kindreds with presumably dominant inheritance of CFM demonstrate the wide phenotypic variability of this condition[3,4].

Previous studies have demonstrated that this pattern of malformations is a condition of etiologic heterogeneity, with genetic and non-genetic risk factors[1,5,6]. Genes involved in embryologic formation of the ears and mandible involve a variety of cellular processes. Variants in transcription factors involved in neural crest cell migration and patterning (TFAP2A, SIX1, SIX5, EYA1, HOXA10, HOXA2), chromatin modifiers (CHD7, KMT2D, KDM6A), growth factors and their receptors (GDF6, FGF3, FGF10, FGFR2, FGFR3), DNA pre-replication complexes (ORC1, ORC4, ORC6, CDC6, CDT1), ribosome assembly (TCOF1, POL1RC, POL1RD), and the spliceosome (EFTUD2, TXNL4A, SF3B4) have been implicated in monogenic syndromes that include malformation of the ears or mandible[7]. Variants in MYT1 have been identified in unrelated individuals with CFM, suggesting a possible role in disease pathogenesis[8,9]. The two largest families described in the literature to date demonstrating dominant inheritance of CFM identified linkage of the trait to chromosomal bands 11q12-13 and 14q32; however, the responsible gene within each linkage interval was not identified at the time of study[10,11]. Despite these advances, the role of rare variants with large effect on disease risk remains unknown. Here, we show a role for haploinsufficient variants in SF3B2 as the most frequent genetic cause of CFM identified to date.

## Results

**Exome sequencing of CFM kindreds**. To identify novel loci for CFM, we performed whole exome ($n = 42$) or whole genome ($n = 104$) sequencing on 138 case-parent trios with sporadic disease, and eight kindreds with multiple affected members. On average, we identified 1.11 coding de novo variants per proband, closely matching expectation and prior experimental results (Table 1)[12]. In comparing the burden of de novo variants in 138 trios with sporadic CFM to expectation, a significant excess of loss of function (LOF) variants was identified in cases (1.7-fold excess, $P = 0.01$). No other variant class demonstrated enrichment in comparison to expectation by chance (Table 1). A single gene, SF3B2 (MIM: 605591), had more than one protein altering de

novo variant, in which we identified de novo LOFs in two distinct probands: a single base deletion at a canonical splice acceptor (c.1780-2delA; Kindred 3) and a frameshift variant (p.D766EfsX4; Kindred 1). The identification of two de novo LOF variants in SF3B2 in a cohort of this size was highly unlikely to occur by chance ($P = 9.6 \times 10^{-7}$; Supplementary Fig. 1), surpassing conservative thresholds for genome-wide significance. In a multiplex kindred, we identified a transmitted LOF in SF3B2 (p.A827RfsX5; Kindred 2) in a proband with bilateral CFM, in whom the variant was inherited from an affected father (Fig. 1; Table 2). Strikingly, SF3B2 lies within the 11q12-13 linkage region (odds in favor of

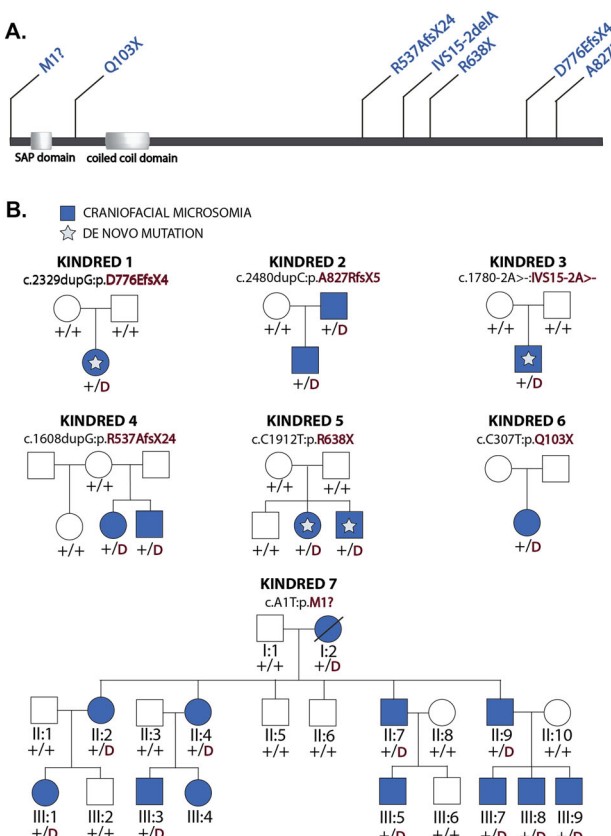

Fig. 1 **SF3B2 variants in pedigrees with craniofacial microsomia. A** Domain structure of SF3B2 with variants identified noted above. Domain annotation based on UniProt accession Q13435. **B** Pedigrees of kindreds with SF3B2 LOF variants. De novo or transmitted LOF variants are indicated above each pedigree, with stars denoting confirmed de novo variants. '+' represents a wild type allele, and 'D' represents the LOF variant in SF3B2 indicated above the pedigree. Individuals with no genotype listed represent those in whom genomic DNA was not available for study.

---

**Table 1 Burden of de novo variants in 138 probands with sporadic CFM.**

| Class | Observed | | Expected | | Enrichment | P value |
|---|---|---|---|---|---|---|
| | # | #/subject | # | #/subject | | |
| All variants | 153 | 1.11 | 154.2 | 1.12 | 0.99 | 0.55 |
| Synonymous | 48 | 0.35 | 43.8 | 0.32 | 1.10 | 0.28 |
| Protein altering | 101 | 0.73 | 110.4 | 0.80 | 0.92 | 0.83 |
| Total missense | 78 | 0.57 | 96.9 | 0.70 | 0.81 | 0.98 |
| Damaging missense | 15 | 0.11 | 18.2 | 0.13 | 0.83 | 0.80 |
| Loss of function (LOF) | 23 | 0.17 | 13.6 | 0.10 | 1.69 | **0.01** |

Bold values represent significant enrichment.
#, number of de novo variants in 138 probands with sporadic CFM; Rate, number of de novo variants per subject; Damaging missense as called by MetaSVM; Loss of function denotes premature termination, frameshift, or splice site variant. P values represent the uncorrected upper tail of the Poisson probability density function (one-sided).

**Table 2 Clinical characteristics of individuals with pathogenic *SF3B2* variants.**

| Proband ID | 1 | 2 | 3 | 4-1 | 4-2 | 5-1 | 5-2 | 6 |
|---|---|---|---|---|---|---|---|---|
| Inheritance | de novo | dominant (pat) | de novo | Unknown[a] | Unknown[a] | de novo | de novo | Unknown[b] |
| Nucleotide change (GenBank NM_006842.2) | c.2329dupG | c.2480dupC | c.1780-2 A > - | c.1608dupG | c.1608dupG | c.1912 C > T | c.1912 C > T | c.307 C > T |
| Amino Acid change | p.D776EfsX4 | p.A827RfsX5 | - | p.R537AfsX24 | p.R537AfsX24 | p.R638X | p.R638X | p. Q103X |
| Gender | Female | Male | Male | Female | Male | Female | Male | Female |
| Age at exam (years) | 17 | 9 | 10 | 12 | 17 | 30 | 28 | 1.5 |
| Weight in Kg (%tile) | 61 (50th–75th) | 38 (90th–98th) | 27 (10th–20th) | 34 (25th) | 82 (90th) | 48 (3rd–10th) | 71 (50th) | 8 (3rd–10th) |
| Height in cm (%tile) | 165 (50th–75th) | 135 (50th–75th) | 135 (20th–30th) | 137 (<3rd) | 156 (<3rd) | 164 (50th) | 178 (50th–75th) | 67 (<3rd) |
| OFC in cm (%tile) | 54 (50th) | 52 (50th) | 52 (20th–40th) | 51 (3rd–10th) | 50 (<3rd) | 54 (25th–50th) | 56 (50th) | 42(<3rd) |
| Facial asymmetry | L mild mandibular hypoplasia | R moderate maxillary, mandibular, and zygomatic hypoplasia | R moderate maxillary and mandibular hypoplasia | L moderate maxillary and mandibular hypoplasia; L orbital dystopia | L mild maxillary and mandibular hypoplasia; L orbital dystopia | L maxillary and mandibular hypoplasia, zygomatic hypoplasia | - | - |
| Coloboma | - | - | - | - | - | - | - | Upper eyelid |
| Lateral oral cleft | - | - | R side | L side | - | Bilateral | - | - |
| Temporomandibular joint | - | Absence on R | - | - | - | - | - | - |
| Ear abnormalities | Duplication of tragus on L side | Duplication of tragus on R side | Bilateral hypoplastic tragus | Bilateral microtia I (EAC atresia on L) | Bilateral microtia I | Bilateral microtia I with R absent tragus, EAC atresia, and multiple sinuses | - | - |
| Skin Tags | L, facial | L complex preauricular (2) | Bilateral, preauricular | L preauricular and facial (multiple) | - | - | L complex preauricular (multiple) | R, preauricular (2) |
| Hearing (by audiologist test) | Normal | R, conductive HL | Normal | L, conductive HL | Normal | Bilateral conductive HL | Congenital perforation of L tympanic membrane | Normal |
| Ophthalmologic anomalies | Myopia | - | - | Ptosis, corrective lenses | Corrective lenses, amblyopia and strabismus | R epibulbar dermoid | - | R exotropia, hypermetropia, severe ptosis |
| Skeletal anomalies | Bilateral cervical ribs (C7) | Cervical rib on L (C7) | Bilateral cervical ribs (C7) | Scoliosis, Hypoplastic 12th ribs and non-rib bearing lumbar vertebral bodies (4), short toes | Bilateral extra flexion crease on thumbs, knee valgus, pes planus | - (No X-ray) | - (No X-ray) | L Bifid thumb (No X-ray) |
| Cardiac anomalies (Echocardiogram) | Absent L pulmonary artery, aberrant L subclavian artery, R sided aortic arch | NP | None | Multiple muscular VSD | None | NP | NP | NP |
| Kidney anomalies (US) | None | None | None | None | None | NP | NP | NP |
| Other birth defects | - | - | - | Submucous CP, Torticollis, anteriorly placed anus, dry skin, thin hair, scant eyebrows | Premature adrenarche, mild 2-3-4 skin syndactyly of digits | - | - | Mild skin syndactyly of toes |
| Neurodevelopment | Normal | Mild NDD (IEP) | Normal | NDD (IEP) | NDD (IEP) | Normal | Normal | Normal |

*OFC occipitofrontal circumference, %tile percentile, R right side, L left side, NS not specified, HL hearing loss, VSD ventricular septal defect, US ultrasound, NDD neurodevelopmental delay, NP not performed, IEP Individualized Education Program, EAC external auditory canal, CP cleft palate.*
aFather not tested.
bParents not tested.

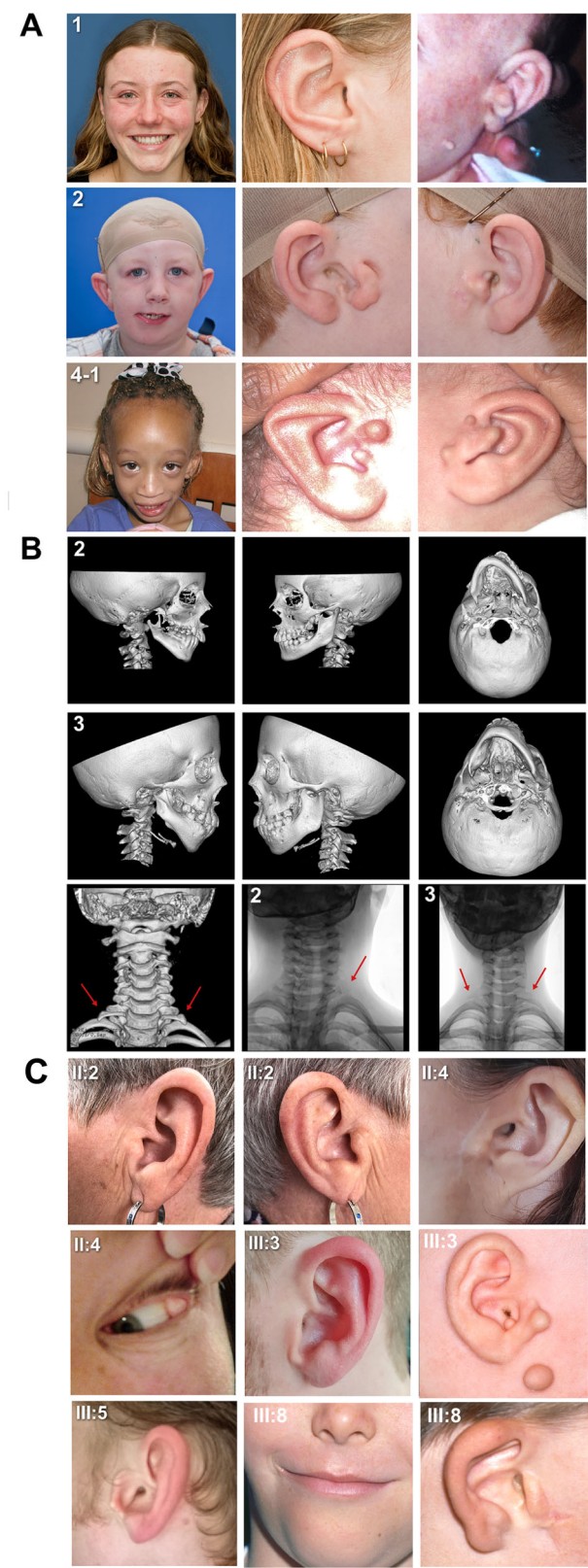

linkage 2048:1) previously identified in the largest reported kindred demonstrating dominant CFM, a family included amongst our multiplex kindreds (Kindred 7)[10,11,13,14]. We performed whole exome sequencing of two distantly related individuals within this pedigree (cousins III:5 and III:8; Fig. 1). A single rare, damaging variant was shared between the two affected cousins within the linkage interval- a single nucleotide substitution causing a startloss in SF3B2 (c.A1T; p.M1?). This variant segregated with 11 affected individuals, with phenotypes including ears with tragal abnormalities, isolated preauricular tags, maxillary and mandibular hypoplasia, epibulbar dermoids, and lateral oral clefts (Fig. 2C, Table 3). In total, we identified novel SF3B2 variants in 4 kindreds. The burden of rare (gnomAD minor allele frequency $< 2 \times 10^{-5}$) LOF variants in SF3B2 in this cohort was

**Fig. 2 Clinical Images of Individuals with *SF3B2* haploinsufficiency. A**. Photographs of three individuals demonstrating the shared phenotype. **Proband 1** (c.2329dupG) demonstrates mandibular hypoplasia on the left, scar from preauricular skin tag excision on the right, and tragal duplication and facial skin tag on the left side prior to removal. **Proband 2** (c.1780-2 A > -) demonstrates maxillary and mandibular hypoplasia on the right side in addition to bilateral tragal abnormalities. **Proband 4-1** (c. 1608dupG) presented with maxillary and mandibular hypoplasia and tragal abnormality on the left side, and bilateral preauricular tags. **B**. Three-dimensional (3D) computed tomography (CT) reconstructions of the craniofacial skeleton of probands 2 and 3. Frontal, lateral, and submental views demonstrate maxillary and mandibular hypoplasia with resulting facial asymmetry. The third row demonstrates, from left to right, 3D CT scans of the cervical spine of **Proband 1** and X-rays of **Proband 2 and 3**. Arrows in each figure indicate the cervical ribs. **C**. Photographs of family members with a shared *SF3B2* startloss variant (p.M1?) demonstrating variable external ear phenotypes. Images of **II:2** demonstrate a right ear markedly smaller than the left. Images of **II:4** demonstrate a tragal abnormality and scar from prior preauricular skin tag removal, in addition to an epibulbar dermoid. **III:3** presented with bilateral tragal abnormalities and tags (right side ear photo from childhood). Images of **III:5** show an underdeveloped tragus. **III:8** had tragal abnormality, scar from tag excision, right lateral oral cleft, and chin deviation due to mandibular hypoplasia.

---

highly enriched in comparison to those amongst more than 250,000 alleles in gnomAD[15] ($P = 3.8 \times 10^{-10}$; Fisher's exact test), and each allele identified in CFM probands was novel (Table 4). *SF3B2* has a pLI score of 1.0, indicating very high intolerance to heterozygous LOF variation, and consequently a high likelihood of phenotypic consequence associated with its mutation. All four probands harboring these *SF3B2* variants presented with ear malformations limited to the tragus, mandibular hypoplasia, preauricular and/or facial tags, and cervical ribs in the three with radiographs available for study (Tables 2, 3; Fig. 2). Together, these findings implicate LOF variants in *SF3B2* as a cause of CFM.

We assessed variant calls from probands for rare, damaging (as called by Meta-SVM) missense variants in *SF3B2*, as well as structural variants encompassing the gene, however none were identified. Analysis of structural variants in the DECIPHER database identified a child with a 1 Mb deletion encompassing *SF3B2* presenting with features of CFM (microtia, preauricular skin tag, other craniofacial dysmorphism)[16,17], suggesting that deletions encompassing the gene could also cause some cases of CFM. No gene had more than one rare, recessive genotype, and no other single gene approached significance in analyzing the burden of rare, damaging alleles in cases. Excluding kindreds with *SF3B2* variants, analysis of de novo mutation burden in our cohort identified a persistently significant excess of de novo LOF mutations in cases (1.6-fold excess; $P = 0.03$, Poisson test; Supplementary Table 1). From the observed excess of de novo LOFs in cases compared to expectation, we estimate that an additional ~5.6% of cases are attributable to these mutations (Methods).

**Replication of haploinsufficient *SF3B2* variants in CFM.** Using the Genematcher platform[18], we identified an additional 5 individuals from 3 kindreds (Kindreds 4,5,6) with heterozygous LOF variants in *SF3B2* (p.R537AfsX34, p.R638X, p.Q103X). These individuals shared the CFM phenotype observed in our cohort, and some demonstrated novel cardiac and limb malformations (Table 2, Fig. 1). In sporadic cases wherein both parents were available for testing, these variants were found to occur de novo in the proband, including one kindred wherein two affected children shared a nonsense variant (p.R638X), however parental samples with confirmed kindship demonstrated no evidence of the variant, indicating gonadal mosaicism (Fig. 1, Table 2). Clinical features of each affected individual, Sanger sequencing traces of identified variants, and a clinical synopsis of each proband are found in Table 2, Supplementary Fig. 2, and Supplementary Note 1, respectively.

**SF3B2 knockdown interferes with neural crest development in *Xenopus laevis*.** The congenital anomalies seen in CFM suggest a disturbance in cranial neural crest (NC) cell formation, which typically migrate into the first and second pharyngeal arches. This paucity of NC cells is thought to underlie the craniofacial phenotype of CFM, particularly the underdevelopment of the mandible and external ear[19,20]. Variants in *SF3B4*, a binding partner of *SF3B2*, cause a craniofacial condition characterized by mandibular and external ear abnormalities known as Nager syndrome (MIM:154400)[21]. Previous work has shown that in an animal model of Nager syndrome, SF3B4 knockdown leads to loss of NC progenitors and subsequent defects in NC-derived craniofacial cartilages[22]. We sought to determine whether the craniofacial anomalies seen in probands with *SF3B2* haploinsufficiency might also be due to defects in NC precursor formation. We first confirmed the expression of *sf3b2* in *Xenopus* embryos by in situ hybrization where *sf3b2* appears to be ubiquitously expressed at all stages examined (Supplementary Figure 3). We then performed microinjection of *SF3B2* translation-blocking morpholino antisense oligonucleotides (SF3B2MO) in *Xenopus* embryos, and analyzed the expression of the NC specific gene *sox10* by in situ hybridization at the neurula stage (stage 15). Unilateral injection of increasing doses of SF3B2MO (10–30 ng) severely affected *sox10* expression in these embryos in a dose dependent manner, while injection of a control MO (CoMO) did not significantly affect the expression of this gene. For the highest dose of SF3B2MO injected (30 ng), 83% of embryos had a reduced or complete loss of *sox10* expression, identifying a dosage sensitive role of SF3B2 in NC progenitor formation ($P < 0.005$, One-way ANOVA using all injection concentrations (10–30ng); Fig. 3A, B). The phenotype of *SF3B2* morphant embryos was efficiently rescued by injection of human *SF3B2* plasmid DNA (10 and 100 pg) in a dose-dependent manner, demonstrating the specificity of the phenotype (Fig. 3A, B). We expanded our characterization of the morphant phenotype by assessing the expression of a broader repertoire of genes expressed in NC (*sox9, tfap2e, snai2*) or neural plate (*sox2*) progenitors. While *sox9* expression was largely unaffected, *tfap2e* and *snai2* were reduced, similar to *sox10*, and the *sox2* expression domain was expanded, consistent with a loss of neural plate border (Fig. 3C, D). At the tailbud stage (stages 25 and 28), when NC cells initiate their migration into the pharyngeal arches, SF3B2 morphant embryos showed a marked decrease in the dorso-ventral extension of the NC streams as visualized by *sox10*, *sox9* and *twist1* expression (Fig. 3E, F). At stage 40, *runx2* is detected in foci of cartilage precursors in the mesenchyme of the pharyngeal arches, and its expression in these domains was severely disrupted in SF3B2MO-injected tadpoles (Fig. 3F, G). We subsequently analyzed stage 45 *Xenopus* tadpoles to evaluate the long-term consequences of SF3B2 depletion on craniofacial development. For the lower doses of SF3B2MO (10 ng and 20 ng), we observed significant craniofacial defects affecting the overall size of the head on the injected side, while CoMO-injected tadpoles (30 ng) were unaffected (Fig. 4A, B). The phenotype was even more pronounced for the higher doses of SF3B2MO (30 ng), however only 35% of these embryos survived to the tadpole stage. Alcian blue staining

**Table 3 Clinical characteristics of individuals from Kindred 7, a family with autosomal dominant CFM and a pathogenic SF3B2 variant (c.A1T:p.M1?).**

| Clinical features | I:2 | II:2 | II:4 | II:7 | II:9 | III:1a | III:3 | III:4 | III:5 | III:7 | III:8 | III:9 |
|---|---|---|---|---|---|---|---|---|---|---|---|---|
| Gender | Female | Female | Female | Male | Male | Female | Male | Female | Male | Male | Male | Male |
| Age at last exam (years) | 60 | 63 | 40 | 36 | 35 | 15 | 23 | 15 | 6 | 11 | 9 | 7 |
| Facial asymmetry | L, mild mandibular hypoplasia | – | L, mild mandibular hypoplasia | – | R, mild mandibular hypoplasia; R, mild | L, mild mandibular hypoplasia | R, mild maxillary and mandibular hypoplasia | – | R, mild mandibular hypoplasia | R, mild mandibular hypoplasia | R, moderate maxillary and mandibular hypoplasia; R, moderate | R, mild mandibular hypoplasia |
| Lateral oral cleft | – | – | – | – | – | – | – | – | – | – | – | – |
| Temporomandibular joint[b] | – | – | L condylar hypoplasia | – | R condylar hypoplasia | – | R condylar hypoplasia | – | R condylar hypoplasia | R condylar hypoplasia | R condylar hypoplasia | R condylar hypoplasia |
| Ear abnormalities | – | R smaller than L | L abnormal tragus, absent lobule | R smaller than L | – | – | R abnormal tragus, absent lobule, EAC stenosis | – | R abnormal tragus, EAC stenosis | R abnormal tragus | R microtia I | – |
| Skin Tags | – | R preauric | L preauric | Bilateral preauric | R preauric | – | R preauric | R, preauric | R preauric | – | R preauric | Bilateral preauric |
| Hearing | Normal (NT) | Normal (NT) | Normal (NT) | Normal (NT) | Normal (NT) | Normal (NT) | Normal (NT) | Normal (NT) | Normal (NT) | Normal (NT) | Normal (NT) | Normal (NT) |
| Ophthalmological anomalies | – | – | L epibulbar dermoid | – | – | – | R epibulbar dermoid | – | – | – | – | – |
| Neurodevelopment | Normal | Normal | Normal | Normal | Normal | Normal | Normal | Normal | Normal | Normal | Normal | Normal |
| Other birth defects[c] | Normal | Normal | Normal | – | – | Normal | Normal | Normal | Normal | – | Normal | Normal |

NT Not Tested, R right, L left, EAC external auditory canal, preauric preauricular.
[a]This individual was assessed as not-affected in the initial paper[11]; reassessment in 2020 revealed left mandibular hypoplasia.
[b]By evaluation of panoramic radiographs.
[c]Weight, height and head circumference were unremarkable (not formally documented). Cardiac, skeletal and kidney anomalies were not assessed with imaging.

**Table 4 De novo and transmitted LOF variants in SF3B2 identified in kindreds with CFM.**

| Gene | NM_006842.2 nucleotide change | Protein impact | gnomAD frequency | pLI |
|---|---|---|---|---|
| SF3B2 | A1T | M1? (startloss) | Novel | 1.00 |
| SF3B2 | C307T | Q103X | Novel | 1.00 |
| SF3B2 | 1608dupG | R537AfsX24 | Novel | 1.00 |
| SF3B2 | 1780-2delA | IVS15-2delA | Novel | 1.00 |
| SF3B2 | C1912T | R638X | Novel | 1.00 |
| SF3B2 | 2329dupG | D776EfsX4 | Novel | 1.00 |
| SF3B2 | 2480dupC | A827RfsX5 | Novel | 1.00 |

pLI probability of LOF intolerance.

revealed that NC-derived cartilages are hypoplastic or missing in SF3B2 morphant tadpoles (Fig. 4C). Altogether, these results indicate that the SF3B2 knockdown phenotype is similar to that observed in SF3B4 knockdown, and consistent with a depletion of NC progenitors contributing to the pathoetiology of CFM[22].

## Discussion

In sum, we identified seven kindreds with LOF variants in SF3B2. These variants span the length of the encoded protein, supporting haploinsufficiency as the mechanism of genetic contribution to disease. Individuals with SF3B2 variants showed phenotypic homogeneity via external ear malformations consistently involving the tragus, and mandibular hypoplasia. These findings suggest that these variants predominantly affect development of pharyngeal arch I, which gives rise to the mandible, ear canal, and the tragus[23]. Targeted knockdown of SF3B2 in Xenopus embryos indicates that in the absence of Sf3b2 function, the formation and subsequent development of cranial NC cells is disrupted, suggesting that NC depletion is a contributing factor to the pathoetiology of CFM. This is consistent with other craniofacial conditions, such as Treacher Collins and Nager syndromes[20,22]. Seventy-seven of the 146 kindreds in our cohort had phenotypes involving more than isolated microtia, and 4 of these harbored LOFs in SF3B2; we estimate that ~5% of CFM is attributable to variants in SF3B2. Notably, 25% of familial cases in our cohort were explained by SF3B2 variants. Analysis of SF3B2 variants in a larger cohort will be necessary to further specify the overall contribution of these mutations to CFM. Highly penetrant, dominant alleles have been found to cause a minor fraction (~5%) of the other congenital craniofacial anomalies that demonstrate similarly high locus heterogeneity, including IRF6 in cleft lip/palate[24,25], and SMAD6 in craniosynostosis[12]. While there were no clear examples of incomplete penetrance in our cohort, one proband (Kindred 8; III:1) evaluated in infancy did not demonstrate features of CFM, however repeat clinical evaluation in adolescence after identification of the SF3B2 variant revealed mandibular hypoplasia. These findings imply that those with SF3B2 LOF mutations identified in infancy with no obvious manifestations of the condition should be assessed closely throughout their development for delayed, potentially mild presentations of CFM.

Lateral oral clefts and epibulbar dermoids, features observed in our probands, are two phenotypes that have classically been used to differentiate CFM from other mandibulofacial dysostoses[19]. The identification of widely variable phenotypes within single kindreds supports the notion that microtia and CFM are part of a single phenotypic spectrum. Further investigation will be necessary to determine the extent to which genetic, environmental, and stochastic factors contribute to the phenotypic heterogeneity observed with SF3B2 haploinsufficiency.

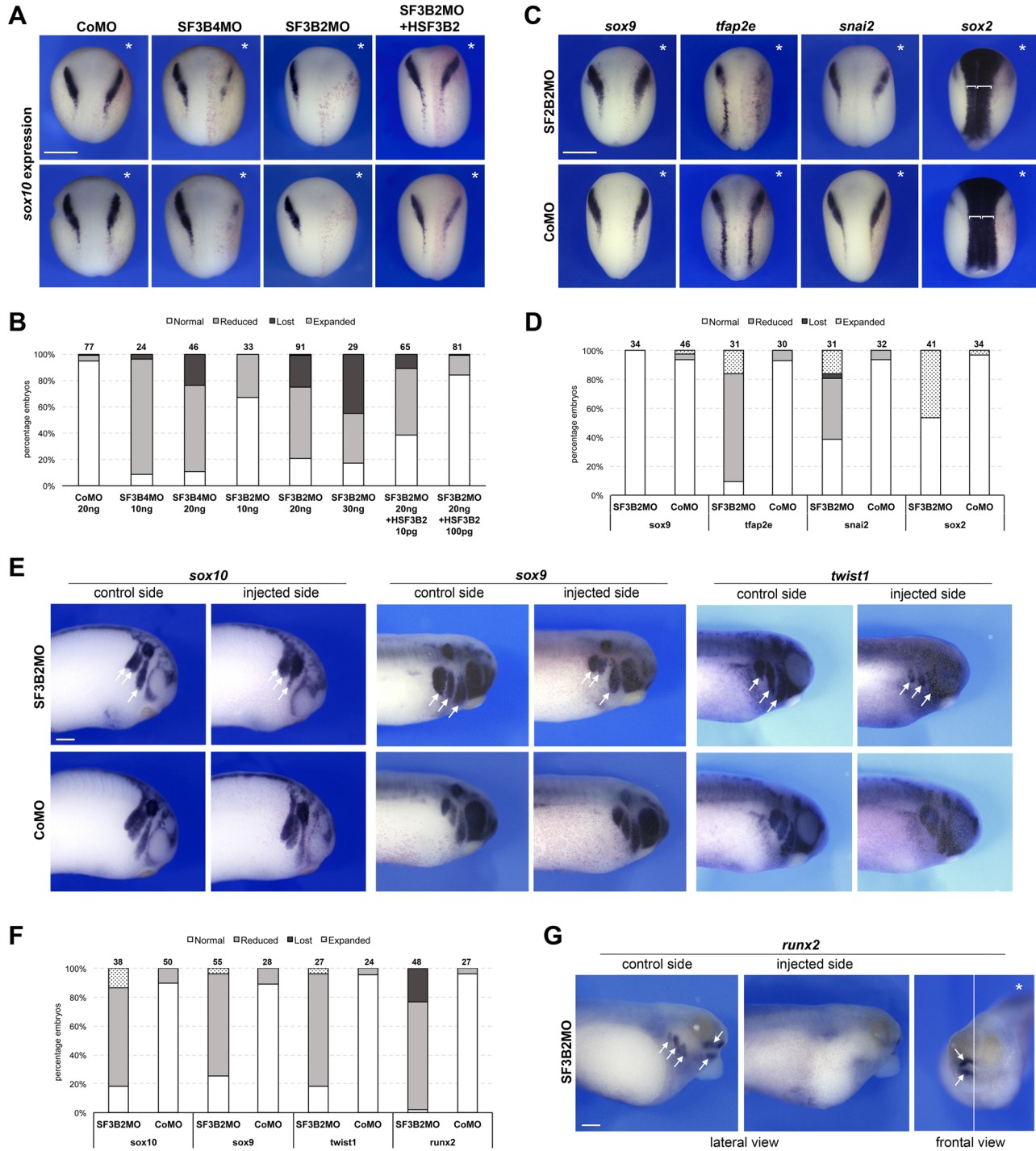

Notably, the often unilateral nature of disease in CFM remains enigmatic. Other common craniofacial anomalies, including cleft lip and craniosynostosis, similarly present with unilateral disease more frequently than bilateral disease. Popular theories to explain this enigma include somatic "second hits" in genes wherein haploinsufficient germline variants predispose to disease, and genetic modifiers of phenotypic presentation. Large families demonstrating substantial phenotypic variability, much like the multiplex kindred presented above (Kindred 7), provide evidence against somatic events causing the disease phenotype in the case of *SF3B2*-associated CFM, as the recurrence of such events in a single gene across multiple generations would be highly unlikely. The potential contribution of somatic mutations in other genes,

as well as genetic modifiers, merit further investigation in CFM and other congenital craniofacial anomalies.

The SF3B complex forms part of the U2 small nuclear ribonucleoprotein complex (U2 snRNP), binding to pre-mRNAs upstream of intron branch sites and anchoring the U2 snRNP to pre-mRNAs. SF3B2 directly interacts with SF3B4 to accomplish these functions. As described above, LOF variants in *SF3B4* cause the acrofacial dysostosis known as Nager syndrome[21]. Patients with Nager syndrome typically have symmetric disease involving the limbs, ears, midface, and mandible. Whereas external ear defects appear frequently in those with *SF3B2* variants, they are restricted to the tragal region and ear canal, and the malar and mandibular hypoplasia that are hallmark features of Nager

**Fig. 3 Sf3b2 knockdown alters neural crest development in *Xenopus* embryos. A** Unilateral injection of increasing doses of SF3B2MO (10-30 ng) interferes with *sox10* expression at the neurula stage, in a manner similar to SF3B4 knockdown. Injection of a control MO (CoMO) did not significantly affect *sox10* expression. The SF3B2 knockdown phenotype is efficiently rescued by injection of human SF3B2 (HSF3B2; 10 pg or 100 pg) plasmid DNA. mRNA encoding the lineage tracer ß -galactosidase was co-injected with the MOs to identify the injected side (punctate red staining). The injected side is indicated by an asterisk. Dorsal view, anterior to top. The two rows show examples of the phenotype for each injection condition. Scale bar, 500 μm. **B** Quantification of the phenotypes. The numbers on the top of each bar indicate the number of embryos analyzed from seven independent experiments. **C** Unilateral injection of SF3B2MO (20 ng) reduces the expression of other neural crest genes at the neurula stage including *tfap2e* and *snai2*, and expanded the neural plate expression domain of *sox2* (white brackets) on the injected side. *sox9* was not affected in these embryos. The injected side is indicated by an asterisk. Dorsal view, anterior to top. Scale bar, 500 μm. **D** Quantification of the phenotypes. The numbers on the top of each bar indicate the number of embryos analyzed from three independent experiments. **E** At the tailbud stage, SF3B2MO (20 ng) injected embryos show a decrease in the length of the NC streams (arrows) as visualized by *sox10* (stage 25), *sox9*, and *twist1* (stage 28) expression. CoMO injection did not affect NC streams formation. Lateral view, dorsal to top, anterior to right. Scale bar, 250 μm. **F** Quantification of the phenotypes. The numbers on the top of each bar indicate the number of embryos analyzed from three independent experiments. **G** At stage 40, the expression of *runx2* in the pharyngeal arches mesenchyme (arrows) is severely downregulated in SF3B2MO-injected tadpoles. The white line indicates the midline (frontal view). Scale bar, 250 μm. The quantification of the *runx2* phenotype is shown in panel **F**. Source data are provided as a source data file.

syndrome appear to be less severe in probands with *SF3B2* variants. Limb malformations were infrequent in probands with *SF3B2*-associated CFM in contrast to the near complete penetrance of limb anomalies in Nager syndrome[21]. Whereas the majority of children with Nager syndrome require craniofacial surgery to maintain airway patency, none of the probands with *SF3B2* haploinsufficiency in this cohort required a tracheostomy or early mandibular surgery for airway compromise.

Interestingly, we identified cervical ribs in three individuals with *SF3B2* haploinsufficiency; the exact frequency in the cohort is unknown since this abnormality is commonly overlooked and we did not have radiographs available for review for the other individuals. Cervical ribs are found in ~1% of the general population, and are usually unilateral. Rib anomalies are frequent in patients with another craniofacial spliceosomopathy, cerebrocosto-mandibular syndrome, caused by variants in *SNRPB*[26]. These findings strengthen the link between spliceosome function and rib patterning in development.

Recent studies have elucidated a role of spliceosomal dysfunction in the retention of poison exons; these highly conserved alternative exons contain a premature termination codon, and are usually spliced out of pre-mRNAs. While found in many genes with significant roles in development and morphogenesis, the exact function of poison exons remains unknown[27]. We identify a significant loss of NC precursors in SF3B2 deficient embryos; it will be interesting to determine whether retention of poison exons via aberrant splicing might contribute to this disease process.

The results define *SF3B2* as a novel gene responsible for CFM (Supplementary Fig. 4), demonstrating dominant inheritance with inter- and intrafamilial phenotypic variability. We recommend adding *SF3B2* to craniofacial genetic panels designed to screen patients with features of mandibulofacial dysostosis. Genetic testing will be particularly useful for CFM patients with positive family history, tragal malformations, or preauricular/facial tags, as the high penetrance of these alleles has strong implications for genetic counseling. Recruiting a larger cohort of patients with *SF3B2* variants will be necessary to delineate the full spectrum of phenotypic consequences associated with its mutation. Given the apparently high degree of locus heterogeneity in CFM, the results suggest that sequencing a substantially larger cohort of CFM cases is likely to reveal other genes and pathways in which de novo mutations contribute to disease risk.

## Methods

**Sample population**. To explore the genetic etiology of CFM, we enrolled families with CFM in our multinational research consortium from 2009 to 2020. Individuals between 0 and 18 years of age were eligible for the study if they met at least one of the following inclusion criteria: (1) microtia or anotia; (2) mandibular hypoplasia and preauricular tag; (3) mandibular hypoplasia and facial tag; (4)

mandibular hypoplasia and epibulbar dermoid; (5) mandibular hypoplasia and lateral oral cleft (e.g. macrostomia); (6) preauricular tag and epibulbar dermoid; (7) preauricular tag and lateral oral cleft; (8) facial tag and epibulbar dermoid; (9) lateral oral cleft and epibulbar dermoid; and (10) consenting parent spoke a language in which they were eligible for consent at their enrolling site. Individuals were excluded if they had an abnormal karyotype or a syndromic diagnosis that involves microtia or underdevelopment of the jaw. Detailed phenotype was ascertained from physical examination performed by a medical geneticist or medical chart abstraction, standardized 2D photos, and parental interview on medical history. Prenatal and family history was collected to account for exposure to known teratogenic substances, and to assess recurrence of the phenotype in the family. Blood or saliva samples were collected from the proband and available parents and affected relatives. DNA was extracted from blood and saliva using standard procedures (Qiagen and Oragene kits). Replication cases were obtained using the GeneMatcher (https://genematcher.org/) or DECIPHER (https://decipher.sanger.ac.uk/) platform.

All human studies were reviewed and approved by the institutional review board (IRB) of the Seattle Children's Hospital and at each of the local institutions' IRBs. An informed consent form was signed by all participants, and explicit written consent was provided for publication of clinical images of the face, which are identifiable. The authors affirm that human research participants have seen and read the material to be published and have provided informed consent for publication of the images in Fig. 2.

**Exome and genome sequencing**. Exome sequencing was performed at either the Northwest Genomics Center (University of Washington) or GENEWIZ using the Nimblegen SeqCap Human Exome v2.0 and Agilent SureSelect Human All Exon capture platforms respectively. For three samples studied at GeneDx, exons were captured using the SureSelect Human All Exon V4 kit, the Clinical Research Exome kit, or the IDT xGen Exome Research Panel v1.0. Whole genome sequencing was performed at the Broad Institute or the Northwest Genomics Center, with sequence data generated on the Illumina HiSeq X platform.

**Variant analyses**. Variants were called using the GATK pipeline (https://gatk.broadinstitute.org/) and annotated using ANNOVAR (http://www.annovar.openbioinformatics.org/), with allele frequencies derived from the gnomAD database (https://gnomad.broadinstitute.org/)[15,28]. De novo variants were called using TrioDeNovo (https://genome.sph.umich.edu/wiki/Triodenovo)[29]. Variants contributing to significant results were validated by bidirectional Sanger sequencing of the proband and both parents. The impact of nonsynonymous variants was predicted using the MetaSVM rank score[30]. Analysis of de novo variant burden in cases was performed in R using denovolyzeR (denovolyzer.org), wherein the observed distribution of variants was compared to the expected Poisson distribution[31,32]. To assess the burden of LOF variants in 145 kindreds with CFM, the total number of LOF alleles identified in each gene in cases was compared to the number identified amongst the median number of alleles across all variant positions (250,396 alleles for *SF3B2*) in gnomAD using Fisher's exact test.

**Contribution of de novo mutations to CFM**. We infer that the number of probands with de novo LOF mutations amongst 136 trios with sporadic CFM without *SF3B2* mutations ($n = 21$) in excess of those expected by chance ($n = 13.4$) represents the number of subjects in whom these mutations confer CFM risk ($n = 7.6$). Comparing this number to the total number of trios considered (7.6/136) yields the fraction of patients in whom these mutations are expected to contribute to disease risk: ~5.6%.

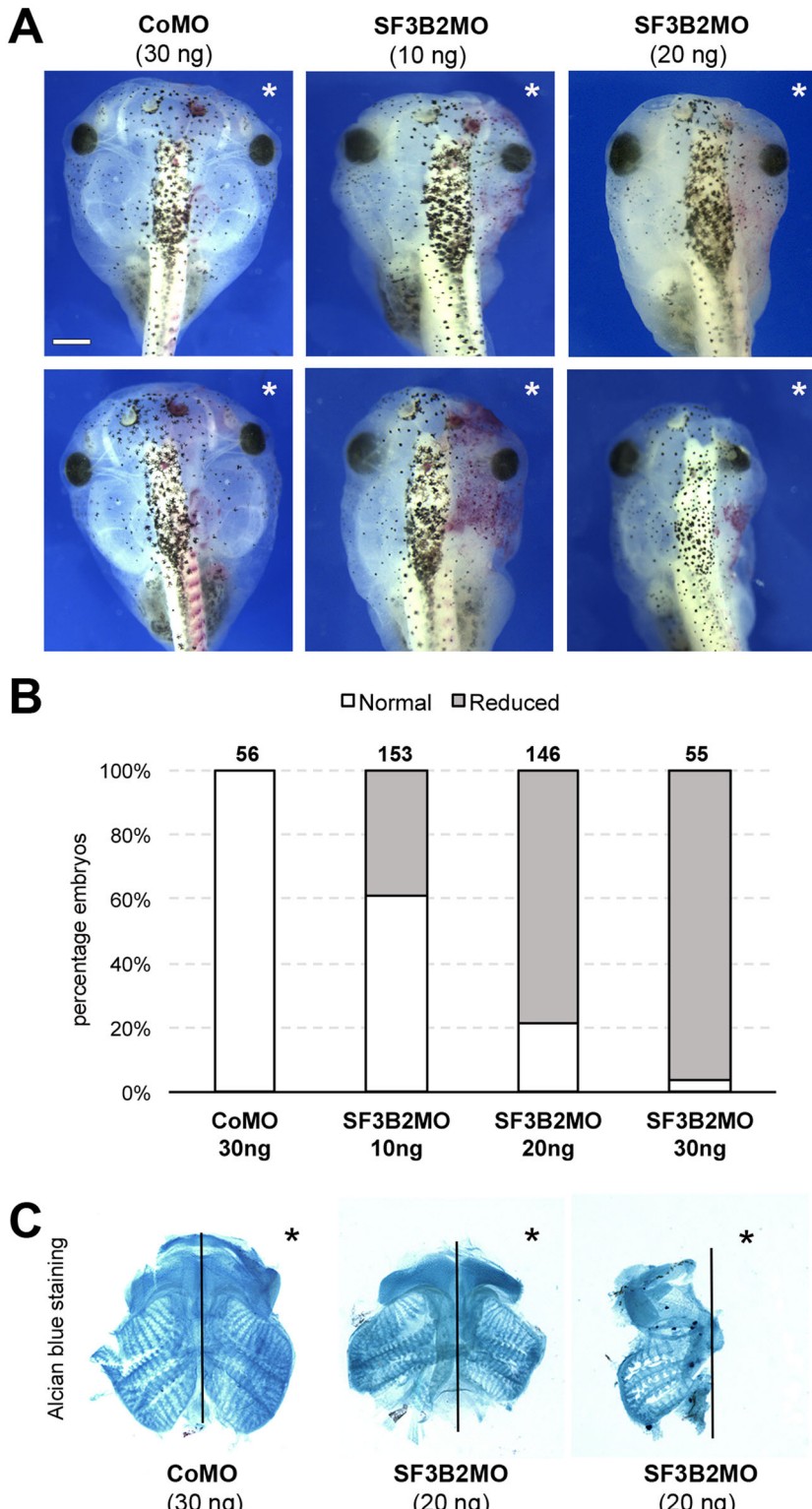

**Fig. 4 Sf3b2 knockdown causes craniofacial defects in *Xenopus* tadpoles. A** Gross morphology of the head of SF3B2MO- and CoMO-injected *Xenopus* tadpoles. The lineage tracer ß -galactosidase was again co-injected, with the injected side indicted by an asterisk. Dorsal views, anterior to top. The two rows show examples of the phenotype for each injection condition. Scale bar, 600 μm. **B** The graph is a quantification of the results from three independent experiments. The number of tadpoles analyzed is indicated on the top of each bar. **C** Alcian blue staining of dissected craniofacial cartilages of CoMO (30 ng) and SF3B2MO (20 ng) injected tadpoles at stage 45. The injected side is indicated by an asterisk. The black lines indicate the midline. Source data are provided as a source data file.

**Morpholino analysis in *Xenopus laevis***. *Xenopus laevis* embryos were raised in 0.1X NAM (Normal Amphibian Medium)[33]. The coding sequence of human SF3B2 (HSF3B2) inserted into pCS2+ expression vector was purchased from GenScript (Piscataway, NJ). Morpholino antisense oligonucleotides (MOs) were purchased from GeneTools (Philomath, OR). Control (CoMO), SF3B4 (SF3B4MO, GCCATAACCTGTGAGGAAAAAGAGC)[22], and two SF3B2 translation blocking MOs (SF3B2MO) targeting each *sf3b2* allo-allele (SF3B2.SMO, CATATCCTCTCT ACTTCCATATAAA; SF3B2.LMO, TTCCGCCATGTTTGCGGTATTTAAG; used at a 1:1 ratio) were injected at the 2-cell stage in one blastomere, along with 500 pg of ß-galactosidase mRNA as a lineage tracer to identify the injected side, and the embryos cultured until stage 15, stage 25, stage 28 or stage 40. For rescue experiments SF3B2MO (20 ng) was coinjected with 10 pg or 100 pg of pCS2+ HSF3B4 plasmid DNA. Prior to in situ hybridization, embryos were fixed in MEMFA (0.1 M MOPS, 2 mM EGTA, 1 mM MgSO$_4$ and 3.7% formaldehyde) and stained for Red-Gal (Research Organics; Cleveland, OH) to visualize the lineage tracer (ß-gal mRNA). Antisense digoxygenin-labeled probes (Genius Kit; Roche, Indianapolis, IN) were synthesized using template cDNA encoding *sox10*, *sox9*, *snai2*, *tfap2e*, *twist1*, *runx2* and *sox2*[34–40]. Whole-mount in situ hybridization was performed[41]. For Alcain blue staining, stage 45 tadpoles were fixed in MEMFA for 1 h, rinsed in H$_2$O, skinned, eviscerated and stained[22]. Oligonucleotides used are described in Supplementary Table 2. All experiments performed were approved by the NYU Institutional Review Board and Institutional Animal Care and Use Committee.

## Data availability

The genetic sequencing data generated in this study have been deposited in the dbGaP database under accession code phs002130.v1.p1. Source data are provided with this paper.

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

## Acknowledgements

We thank the patients and parents for their participation in this research. We thank Dr. Jeff Otjen from Seattle Children's Hospital, who reviewed the radiological images, and Erin Torti of GeneDx for connecting our team with referring clinicians whose patients had *SF3B2* variants. The research leading to these results has received funding from NIH-NIDCD R00DC011282 (to D.L.), NIH-NIDCR RC1 DE 020270 (to C.H.), NIH-NICHD X01 HL140518-01 (to D.L.), NIH-NICDR U01 DE025862 (to D.L. and C.H.), NIH-R01DE025468 (to J.P.S.J.), and the Jean Renny Endowment for Cranio-facial Research (to M.L.C.). We would like to acknowledge Dr. David Chandler of the Australian Genome Research Facility (Node Manager, Perth Australian Genome Research Facility) for his work in identifying the first linkage of the Western Australia CFM family to chromosome 11 in 2001, and Stephanie Hedges for her clinical con-tributions. Whole genome sequence analysis was provided by the University of Washington Center for Mendelian Genomics (UW-CMG) and was funded by NHGRI and NHLBI grants UM1 HG006493 and U24 HG008956. The content is solely the responsibility of the authors and does not necessarily represent the official views of the National Institutes of Health.

## Author contributions

D.L. and C.H. conceived the overall project and led the phenotypic data harmonization. A.T.T., A.E.T., J.A.G., D.A.S., J.G., S.L.S., N.G.L., C.A.W., and D.V.L. performed genetic analyses and interpretation. A.T.T., N.K., D.C., R.E.S., S.J.D., C.H., I.Z., H.P., G.L.P.H., P. H.V., M.M.D.R., M.R.D., A.F.D., L.M., S.V.M., A.J., A.V.H., M.L.C., J.S., and M.R.D. acquired clinical data, collected specimens and contributed clinical evaluations; C.G. and J.P.S.J. performed functional analyses/molecular characterization in *Xenopus laevis*; A.T. T., C.G., J.P.S.J. and D.V.L. wrote the manuscript with input from all authors.

## Competing interests

Rhonda E. Schnur and Sureni V. Mullegama are employees of GeneDx. The remaining authors declare no competing interest.

## Additional information

[1]Hansjorg Wyss Department of Plastic and Reconstructive Surgery, NYU Langone Medical Center, New York, NY, USA. [2]Department of Molecular Pathobiology, New York University College of Dentistry, New York, NY, USA. [3]Department of Pediatrics, Division of Craniofacial Medicine, University of Washington, Seattle, WA, USA. [4]Center for Developmental Biology and Regenerative Medicine, Seattle Children's Research Institute, Seattle, WA, USA. [5]Division of Clinical and Metabolic Genetics, Department of Pediatrics, The Hospital for Sick Children, University of Toronto, Toronto, ON, Canada. [6]The Prenatal Diagnosis and Medical Genetics Program, Department of Obstetrics and Gynecology, Mount Sinai Hospital, University of Toronto, Toronto, ON, Canada. [7]Department of Diagnostic Genomics, Path West Laboratory Medicine, QEII Medical Centre, Hospital Avenue, Nedlands, WA, Australia. [8]Genetics Program, Peterborough Regional Health Centre, Peterborough, ON, Canada. [9]Department of Otolaryngology/Head and Neck Surgery, University of North Carolina, Chapel Hill, NC, USA. [10]Hospital Edgardo Rebagliati Martins, EsSalud, Lima, Peru. [11]Genetic Services of Western Australia, King Edward Memorial Hospital, Perth, WA, Australia. [12]Pontificia Universidad Javeriana and Centro Médico Imbanaco, Cali, Colombia. [13]Division of Plastic and Maxillofacial Surgery, Children's Hospital Los Angeles, Los Angeles, CA, USA. [14]Department of Pediatrics, London Health Sciences Centre, Division of Medical Genetics, Western University, London, ON, Canada. [15]Neurogenetic Diseases Group, Harry Perkins Institute of Medical Research and Centre for Medical Research, University of Western Australia, Nedlands, WA, Australia. [16]Division of Plastic and Reconstructive Surgery, Children's Hospital of Philadelphia, Philadelphia, PA, USA. [17]GeneDx, Gaithersburg, MD, USA. [18]Universidad Icesi and Fundacion Clinica Valle del Lili, Cali, Colombia. [19]Clinica Comfamiliar Risaralda, Pereira, Colombia. [20]Dept of Pediatrics, Cooper Medical School of Rowan University; Division of Genetics, Cooper University Health Care, Camden, NJ, USA. [21]Perth Children's Hospital, Nedlands, WA, Australia. [22]Human Genomics Institute, Pontificia Universidad Javeriana, Bogotá, Colombia. [23]Hospital Universitario San Ignacio, Bogotá, Colombia. A list of members and their affiliations appears in Supplementary Information. ✉email: andrew.timberlake@nyumc.org; luquetti@uw.edu

