## [Peer Review File · Nature Communications]

Reviewer #1 (Remarks to the Author):

The authors present an well designed and executes study to identify the genetic basis or craniofacial microsomia (CFM - the etiology of which had been mostly elusive except for the identification of few rare CNVs. The leading hypothesis had either been non-genetics, mosaicism of very high levels of genetic heterogeneity. Using a large discovery cohort of 146 kindreds with sporadic(n=138) or familial (n=8) CFM, a genome wide sequencing approach was used. Using genematcher a validation cohort was also recruited.

The main findings that LOF SF3B2 variants is associated is a cause of CFM is well justified with 1) recurrent both denovo variants in sporadic as well as segregating variants in familial cases) 2) LOF is also reported mechanism in SF3B4 pathogenic variants in Nager syndrome), 3) statistical evidence of an overrepresentation of LOF in the cohort) 4) zebrafish data showing depletion of neural crest precursors in SF3B2 morpholinos.

My comments do not challenge the main conclusions of the paper, however at times the paper is hard to follow and I believe some some of the conclusions merit revisiting.

1. The introductory paragraphs refers to 20 cases from 7 kindreds. Please confirm the number of cases as fig 1 and table 2 seem to total 21?
2. I would highly recommend a standardized kindred and case naming and perhaps one that differentiates discovery cohort cases (from the 146 initial cohort) vs confirmation cases recruited from decipher. The test only refers to cases and kindreds by mutation, and it becomes difficult to link test descriptions with specific kindreds and cases in the tables and figures.
3. Please confirm or defend the conclusion that 5% of sporadic CFM is due to SF3B2? I believe 2 cases of the original 138 sporadic cases had denovo LOF variants which is 1.4%. The additional cases were recruited from Genematcher but do not represent a population cohort and should not therefore be included in population frequency calculations. It is clearly that a higher proportion of familial cases are due to SF3B4, 25% of the initial cohort.
4. I find the introductory paragraph would benefit from further data. Of the 20 (or 21) cases over 50% are from a single large family. I would prefer to see a summary of the frequency of SFB2 variants in sporadic and familial cases. Between the title and the introductory paragraph, a casual reader could easily miss understand that although SF3B2 is without a doubt a cause of CFM, the majority of cases remain unexplained.
5. Table 1 presents intriguing evidence of an excess of LOF variants in the discovery cohort. Does this excess of LOF variants persist if the SF3B2 variants are removed? The results of this analysis might provide insight into the etiology of the non-SF3B2 cases ie if the excess of cases persists, it would suggest that additional LOF non-recurrent LOF variants are responsible for CFM. If the excess LOF was entirely due to SF3B2, then what does that suggest?
6. The clinical tables and clinical figures show nicely the range of clinical phenotypes in the SF3B2 cases, but also quite convincingly that the cases are CFM "spectrum" and not a unique subset / novel syndrome within the cohort.

Reviewer #2 (Remarks to the Author):

Given the high incidence of hemifacial microsomia there is considerable interest in understanding

the etiology and pathogenesis of this fairly common craniofacial disorder.

Through whole genome and whole exome sequencing of nearly 150 case parent trios with the sporadic condition and 8 families with multiple heritably affected members, variants in SF3B2 were identified as being significantly associated with the disorder.

The work can make an important contribution to the field. The pedigree analyses and human phenotyping appear to be detailed and well documented. But the standard in the field is to pair human genetic data with functional cell and animal analyses to really prove mutations in SF3B2 are causative and underpin the disorder.

In this regard there is a disconnect between the human phenotype and animal studies. The animal studies performed just aren't sufficient to validate SF3B2 being causative of hemifacial microsomia, not least because the morpholino knockdown studies don't recapitulate any aspects of the human phenotype. The loss of Sox10 expression, a gene important for neuroglial differentiation and a smaller head doesn't mean there is an absence of neural crest cells as neural crest cells can migrate in the absence of Sox10. More genes need to be examined to rule out the possibility of a neural crest cell fate switch and therefore it's critical to examine not only a few other neural crest cell markers (Sox9, Tfap2) but also some early differentiation markers (Sox9, Col2a1, Runx2) and later differentiation markers (cartilage and bone with alizarin red/alcian blue). This is the only way to show that there are long term differentiation defects which mimic the human phenotype. The reason for the speculative absence of neural crest cells should also be explored. Is it due to extensive cell death which might explain the smaller head? I expect based on the morpholino technique that apoptosis will be widespread. Nonetheless this should be explored as well as whether p53 is activated in association with the apoptosis similar to other spliceosomopathy disorders.

Expression data demonstrating where SF3B2 is active during early craniofacial development needs to be provided, particularly with some evidence showing expression in neural crest cells.

Lastly, some minimal studies should be performed either in cell lines or in the knockdown embryos to show that alternative splicing is globally affected by SF3B2 loss-of-function. It would be ideal to do this assay in embryos..

The human p The animal embryo is important in validating the well done, but the animal studies are limited and don't adequately support the conclusions of a pathogenic functional link between SF3B2 in neural crest cells, alternative splicing and abnormal craniofacial development.

Without this accompanying data, the human phenotypic analyses come across as descriptive detailed case studies without any proof for their genetic, cellular and development origins.

Reviewer #3 (Remarks to the Author):

This is a meticulously presented report of the identification of the SF3B2 gene as having both familial and sporadic mutations that can present with the phenotype of craniofacial microsomia a classic disorder of dysmorphology that has been a challenging nut to crack molecularly. They outline the strategy and results for this finding in a lucid, linear fashion and the findings that haploinsufficiency

of SF3B2 can cause CFM are compelling with de novo mutations, mutations transmitted as autosomal dominants (including one in a large pedigree with classic features) and even a case of likely gonadal mosaicism in two affected children with unaffected (and confirmed paternity) parents. Some specific concerns about the work include:

The intro states that “haploinsufficient variants in SF3B2 as the most prevalent genetic cause of CFM” but this should be modified to state that of those identified it is the most common as it still only explains 5% of CFM overall. If there are any examples of non-penetrance that come from the autosomal dominant families they have studied it would be useful to include this information for future genetic counseling protocols. They do comment in table 3 that one individual had their phenotype changed in 2020 on further examination so if that was mild and especially if post sequence results that is important for analyses that will be done on infants. This pedigree also has both L and R sided affecteds which is relevant to the comment below on sidedness. Similarly are there any examples of chromosomal deletions (or duplications) of the SF3B2 locus that could further support the haploinsufficiency model? Next in supplementary figure 2 one family with the IVS15-2 label has Sanger sequencing that appears to show that the mother has the same mutation as the proband but it is not boxed or commented on. I could also not align that table and its labels with table 1 in the main text. Please clarify what is going on here as in is that mother the mosaic they refer to? One of the dominant families? The labeling should be consistent.

See below for the xenopus work, but do they have any hypotheses why, even in the dominant families, this disorder is most commonly unilateral? This is one of its truly distinguishing features of CFM and is there data from mouse Kos or even mouse expression studies that show, for example, asymmetric expression or is SF3B2 in a pathway that includes other genes that have sidedness in their expression. In the discussion there was no note of how their prior work on how MYT1 fits into this picture both in terms of its clinical contributions (for example do any of the patients with SF3B2 variants also have MYT1 variants or is MYT1 in any pathways common with SF3B2? Since the overall numbers explained by SF3B2 (5%) are modest helping to focus the search next for future genetic contributions could be a useful discussion point.

They also present supportive data from an animal model (xenopus) for this finding. Here the story is perhaps less rigorous, so at least the authors might do some specificity controls for their antisense knockdown (see below, rescue and show expression of the gene in the neural crest or ubiquitous). Showing more mechanism in the xenopus models would improve this section. Although, they show the sox10-inhibition phenotype is very similar to that of Sf3b4 knockdown (here and in a previous paper); and since the syndromes seems so similar and the proteins are proposed to act together in the U2 snRNP complex, it's unclear whether this represents a significant advancement. Also, unfortunately the frog work probably can't speak much to the unilaterality of the human phenotype since their effects are all from unilateral injections. Overall these assays are useful for working out mechanisms of the genes involved and for finding new candidates. However, the extent that Sf3b2/4-depletion of neural crest cell populations precisely mimics the human disease etiology (or whether these genes are also involved in different/more differentiated cell types later on) would require tissue-specific and/or temporal genetic ablation.

The specificity and efficacy of the antisense morpholino oligo (MO) was not tested for by the usual methods: 1) showing the MO can block in vitro translation of the RNA, 2) rescue of the phenotype with injected mRNA that is not bound by the MO (can be the human homologue) or using two non-overlapping MOs (not preferred—only if the injected RNA causes abnormalities). Also, the authors didn't show that the gene is even expressed in the tissue of interest (ie, neural crest). The labs involved here can do all those experiments and have done so in prior work on SF3B4 (involved in the

similar Nager and Rodriguez Syndromes, Ref. 20 in the submitted paper, Devotta et al., 2016), which could explain the apparently stronger effect of Sf3b2 depletion. The authors may also consider testing putative loss-of-function variants (ie, point mutants or small indels) by showing failure to rescue the MO depletion (as in Devotta et al). Since both the Sf3b2/4 genes are in the U2 snRNP complex, it would be nice to see some more parallel studies to see how similar the phenotypes are (Sf3b4 depletion suggested increased apoptosis as a possible mechanism for neural crest depletion). The structural model seems to suggest that b2 might link b4 to the complex (Sun, C. The SF3b complex: splicing and beyond. Cell. Mol. Life Sci. 77, 3583–3595 (2020). <https://doi.org/10.1007/s00018-020-03493-z>), so the b2 depletion might be stronger (maybe explaining the dominant inheritance?). Many labs also test putative loss-of-function variants by showing failure to rescue the MO depletion (also in Devotta et al).

Other xenopus issues include:

- The protein domain diagram could be modified to include proline-rich and glutamine-rich regions in the N- and C-terminal regions, respectively (and other relevant information from the NCBI protein entry).
- In two instances on p.5 the authors refer to proliferation defects in depleted embryos; studies on proliferation are not presented and thus these statements are not supported by the data (also, the similar phenotypes of Sf3b4-depletions were linked to increased apoptosis, not proliferation, in a previous paper (Devotta, et al., 2016).
- Similarly, the section on 'poison exons' seems speculative, especially since other data from the same paper on Sf3b4 did not show indications of splicing defects, at least on selected neural crest genes.

Minor issues:

The phrase "linkage of the trait to chromosomes 11q12-13 and 14q32 " might reframe chromosome to chromosomal bands.

Missing word, line 5 Introductory paragraph: ". . .nuclear ribonucleoprotein ^, in probands. Twenty . . ." ; ^ complex

Xenopus gene names and symbols are lower case and italics by community convention (<http://www.xenbase.org/entry/>); sf3b2

Overall this is a compelling report that mutations, both familial and de novo, can cause CFM in about 5% of cases, that the SF3B2 gene involved is a logical candidate given its role in neural crest, its paralogs role in Nager S and its interactions with other relevant developmental pathways. It has been carried out carefully, with a very large sample size for a rare disorder and is a quantum advance in our biological understanding of CFM and has important immediate clinical implications for genetic counseling and opens new doors to future biological understanding.

Reviewed by
Jeff Murray
Doug Houston

REVIEWER COMMENTS

Reviewer #1 (Remarks to the Author):

The authors present an well designed and executes study to identify the genetic basis or craniofacial microsomia (CFM - the etiology of which had been mostly elusive except for the identification of few rare CNVs. The leading hypothesis had either been non-genetics, mosaicism of very high levels of genetic heterogeneity. Using a large discovery cohort of 146 kindreds with sporadic(n=138) or familial (n=8) CFM, a genome wide sequencing approach was used. Using genematcher a validation cohort was also recruited.

The main findings that LOF SF3B2 variants is associated is a cause of CFM is well justified with 1) recurrent both de novo variants in sporadic as well as segregating variants in familial cases) 2) LOF is also reported mechanism in SF3B4 pathogenic variants in Nager syndrome), 3) statistical evidence of an overrepresentation of LOF in the cohort) 4) zebrafish data showing depletion of neural crest precursors in SF3B2 morpholinos.

We thank this reviewer for your comments; as you note, our primary aim was to employ a genome wide discovery approach to identify novel genetic causes of craniofacial microsomia.

My comments do not challenge the main conclusions of the paper, however at times the paper is hard to follow and I believe some of the conclusions merit revisiting.

1. The introductory paragraphs refers to 20 cases from 7 kindreds. Please confirm the number of cases as fig 1 and table 2 seem to total 21?

As you note, there are 21 cases across all kindreds, however one affected individual, namely III:4 in the multiplex kindred, was not interested in providing a genetic sample for analysis. While we have phenotypic data for this individual, we can not state with complete certainty that this individual harbors the SF3B2 LOF variant identified in this kindred, and thus this person was not included in our total count of individuals with SF3B2 LOFs identified.

2. I would highly recommend a standardized kindred and case naming and perhaps one that differentiates discovery cohort cases (from the 146 initial cohort) vs confirmation cases recruited from decipher. The test only refers to cases and kindreds by mutation, and it becomes difficult to link test descriptions with specific kindreds and cases in the tables and figures.

This is an excellent suggestion. In Table 2, the top row is “Proband ID” in which we provide the kindred/proband ID’s. These correspond to the ID’s affixed to the clinical images as well. We have added the Kindred ID to the pedigree image in order to make this distinction clearer, and referenced the Kindred ID when introduced in the text in order to provide further clarification.

3. Please confirm or defend the conclusion that 5% of sporadic CFM is due to SF3B2? I believe 2 cases of the original 138 sporadic cases had de novo LOF variants which is 1.4%. The additional cases were recruited from Genematcher but do not represent a population cohort and should not therefore be included in population frequency calculations. It is clearly that a higher proportion of familial cases are due to SF3B4, 25% of the initial cohort.

As you state, cases ascertained via Genematcher do not represent a population cohort and are not included in this calculation. In the discussion section of our manuscript, we describe that 76

of the 146 kindreds in our cohort had phenotypes involving more than isolated microtia, and 4 of these harbored LOFs in *SF3B2*, hence our estimate that ~5% of CFM is attributable to variants in *SF3B2*. We further clarify in our revision: “Analysis of *SF3B2* variants in a larger cohort will be necessary to further specify the overall contribution of these mutations to CFM.” As we note in the discussion, the most prevalent dominant alleles causing other congenital craniofacial anomalies that demonstrate similarly high locus heterogeneity, namely *IRF6* in cleft lip/palate and *SMAD6* in craniosynostosis, are also found in ~5% of cases.

4. I find the introductory paragraph would benefit from further data. Of the 20 (or 21) cases over 50% are from a single large family. I would prefer to see a summary of the frequency of *SF3B2* variants in sporadic and familial cases. Between the title and the introductory paragraph, a casual reader could easily miss understand that although *SF3B2* is without a doubt a cause of CFM, the majority of cases remain unexplained.

This is an excellent point- to further clarify this, we have changed our statement on prevalence in the introductory paragraph to: “The results establish haploinsufficient variants in *SF3B2* as the most prevalent genetic cause of CFM, explaining ~3% of sporadic and 25% of familial cases.” These numbers are derived 2/67 sporadic cases with features of CFM that extend beyond isolated microtia, and 2/8 familial cases. As described above, we now elaborate in the discussion that refining this estimate will require analysis of *SF3B2* variants in a larger cohort.

5. Table 1 presents intriguing evidence of an excess of LOF variants in the discovery cohort. Does this excess of LOF variants persist if the *SF3B2* variants are removed? The results of this analysis might provide insight into the etiology of the non-*SF3B2* cases if the excess of cases persists, it would suggest that additional LOF non-recurrent LOF variants are responsible for CFM. If the excess LOF was entirely due to *SF3B2*, then what does that suggest?

When excluding the 4 kindreds in which *SF3B2* LOF mutations were identified in the discovery cohort, there is still a significant excess of LOF mutations in this cohort. 1.6-fold enrichment, $P=0.03$. The results demonstrate significant locus heterogeneity for CFM, consistent with these findings. To address this, we have added to the results section that when excluding *SF3B2* kindreds, a significant excess of *de novo* LOFs persists, and we provide an estimate that an additional 5.4% of cases are likely to be explained by *de novo* LOFs, with the method used to calculate this estimate described in Methods. We have also added to the concluding paragraph this results implies that sequencing a substantially larger cohort of CFM cases is likely to reveal other genes or pathways in which *de novo* LOF mutations contribute to disease risk.

6. The clinical tables and clinical figures show nicely the range of clinical phenotypes in the *SF3B2* cases, but also quite convincingly that the cases are CFM "spectrum" and not a unique subset / novel syndrome within the cohort.

We agree that this is an important phenotypic distinction, which we highlight in the discussion. These findings lend support to the notion that microtia/CFM/OAVS/Goldenhar syndrome represent a phenotypic spectrum, which has both clinical and genetic implications. Phenotypic modifiers of *SF3B2* haploinsufficiency will be an interesting avenue of investigation in the future. We have added to the discussion section several notes on features common amongst cases with *SF3B2* mutations, and distinctions in comparison to those with *SF3B4* mutations.

Reviewer #2 (Remarks to the Author):

Given the high incidence of hemifacial microsomia there is considerable interest in understanding the etiology and pathogenesis of this fairly common craniofacial disorder.

Through whole genome and whole exome sequencing of nearly 150 case parent trios with the sporadic condition and 8 families with multiple heritably affected members, variants in *SF3B2* were identified as being significantly associated with the disorder.

The work can make an important contribution to the field. The pedigree analyses and human phenotyping appear to be detailed and well documented. But the standard in the field is to pair human genetic data with functional cell and animal analyses to really prove mutations in *SF3B2* are causative and underpin the disorder.

We thank this reviewer for the detailed and thoughtful comments on our manuscript. They have been tremendously helpful in improving the manuscript, and providing further insight into how *SF3B2* mutations might cause CFM. You note, however, that the “standard in the field” is to pair human genetic data with functional cell and animal analyses to prove that mutations are causative in a disease. This is very true when gene “matching” platforms are used to assemble a cohort of individuals with a given phenotype which identifies a concordant phenotype, which certainly requires a further degree of investigation to establish causality of those variants in the disease studied. This manuscript is first and foremost a computational analysis of the genomic data generated by sequencing our cohort, using a completely unbiased approach to gene discovery that has been employed similarly to identify novel disease loci in dozens of other phenotypes. I must note that we identified *SF3B2* in this manuscript via a genome-wide significant burden of *de novo* mutations, a *genome-wide* significant burden of LOF mutations in cases, and a significant LOD score in a single kindred- each of these three methods of gene identification is generally accepted in identifying novel disease loci, and we have done all three with highly significant results in this paper, unequivocally identifying mutations in *SF3B2* as a cause of CFM.

In this regard there is a disconnect between the human phenotype and animal studies. The animal studies performed just aren't sufficient to validate *SF3B2* being causative of hemifacial microsomia, not least because the morpholino knockdown studies don't recapitulate any aspects of the human phenotype. The loss of Sox10 expression, a gene important for neuroglial differentiation and a smaller head doesn't mean there is an absence of neural crest cells as neural crest cells can migrate in the absence of Sox10. More genes need to be examined to rule out the possibility of a neural crest cell fate switch and therefore its critical to examine not only a few other neural crest cell markers (Sox9, Tfp2) but also some early differentiation markers (Sox9, Col2a1, Runx2) and later differentiation markers (cartilage and bone with alizarin red/alcian blue). This is the only way to show that there are long term differentiation defects which mimic the human phenotype.

We agree with the reviewer, and we have significantly developed this aspect of the manuscript. We have expanded the range of neural crest markers analyzed to include genes expressed in the pre-migratory (*snai2*, *sox9* and *tfap2e*) as well as migratory (*sox9*, *sox10* and *twist1*) neural crest. Post neural crest cell migration we have analyzed the expression of *runx2* in the developing pharyngeal arch mesenchyme as a marker of cartilage precursors. We have also performed alcian blue staining to document the long-term consequences of *Sf3b2* depletion on neural crest derived craniofacial cartilages. Our results show that similar to *sox10*, the expression of *tfap2e* and *snai2* was downregulated at stage 15, while the expression of *sox9* was largely unaffected (Fig 3C,D). At the tailbud stage (stage 25-28) as neural crest cells migrate in the pharyngeal arches, most *Sf3b2*-depleted embryos exhibited a marked reduction in the dorso-ventral extension of the neural crest

streams as visualized by *sox10*, *sox9* and *twist1* expression (Fig 3E,F). At stage 40, morphant tadpoles had reduced *runx2* expression in the pharyngeal arches consistent with a decreased number of cartilage precursors (Fig 3F,G). At stage 45, these animals exhibited severe craniofacial defects, with hypoplastic or missing cartilages (Fig 4). Altogether, these results point to defects in cranial neural crest cells formation as the likely cause for CFM associated with *SF3B2* haploinsufficiency.

The reason for the speculative absence of neural crest cells should also be explored. Is it due to extensive cell death which might explain the smaller head? I expect based on the morpholino technique that apoptosis will be widespread. Nonetheless this should be explored as well as whether p53 is activated in association with the apoptosis similar to other spliceosomopathy disorders.

While we agree that this will be quite interesting to explore, we feel that this is outside of the scope of the current manuscript. Further studies will investigate the underlying mechanisms contributing to the phenotype.

Expression data demonstrating where *SF3B2* is active during early craniofacial development needs to be provided, particularly with some evidence showing expression in neural crest cells.

We have analyzed the expression of *sf3b2* by in situ hybridization and show that *sf3b2* is ubiquitously expressed in the embryo at all stages examined, which is not unexpected considering its global role as a splicing factor. The data are included in Supplementary Figure 1.

Lastly, some minimal studies should be performed either in cell lines or in the knockdown embryos to show that alternative splicing is globally affected by *SF3B2* loss-of-function. It would be ideal to do this assay in embryos.

While we agree that identifying which transcripts are alternatively spliced in normal and mutant embryos would be interesting follow-up studies, we feel that this is beyond the scope of the current manuscript.

The human p The animal embryo is important in validating the well done, but the animal studies are limited and don't adequately support the conclusions of a pathogenic functional link between *SF3B2* in neural crest cells, alternative splicing and abnormal craniofacial development.

Without this accompanying data, the human phenotypic analyses come across as descriptive detailed case studies without any proof for their genetic, cellular and development origins.

We have further developed the animal studies to include the developmental expression of *sf3b2*, rescue of the morphant phenotype by injection of human *SF3B2*, analyze of several neural crest genes at different developmental stage and characterize the long-term defects of *Sf3b2* depletion on neural crest derived craniofacial cartilages. Altogether, the results point to defects in neural crest formation as a consequence of *SF3B2* haploinsufficiency. We believe that this adds significantly to the manuscript's connection of *SF3B2* mutations to CFM via neural crest defects.

Reviewer #3 (Remarks to the Author):

This is a meticulously presented report of the identification of the *SF3B2* gene as having both familial and sporadic mutations that can present with the phenotype of craniofacial microsomia a classic disorder of dysmorphology that has been a challenging nut to crack molecularly. They outline the strategy and results

for this finding in a lucid, linear fashion and the findings that haploinsufficiency of SF3B2 can cause CFM are compelling with de novo mutations, mutations transmitted as autosomal dominants (including one in a large pedigree with classic features) and even a case of likely gonadal mosaicism in two affected children with unaffected (and confirmed paternity) parents.

We thank this reviewer for the thoughtful comments.

Some specific concerns about the work include:

The intro states that “haploinsufficient variants in SF3B2 as the most prevalent genetic cause of CFM ” but this should be modified to state that of those identified it is the most common as it still only explains 5% of CFM overall.

This is an excellent point raised by multiple reviewers. To further clarify, we have changed our statement on prevalence in the introductory paragraph to: “The results establish haploinsufficient variants in *SF3B2* as the most prevalent genetic cause of CFM, explaining ~3% of sporadic and 25% of familial cases.” We also have added further comment in the discussion regarding the apparent locus heterogeneity yet residual significant burden of *de novo* LOF mutations when excluding probands with SF3B2 mutations, implying that sequencing a substantially larger cohort will be useful in identifying novel loci that might explain the remaining 95% of cases without known genetic causes, as the excess of *de novo* LOFs beyond those in *SF3B2* are expected to contribute to an additional ~5.4% of sporadic cases. An explanation of this calculation has been added to the Methods section, with associated data added as Supplementary Table 1.

If there are any examples of non-penetrance that come from the autosomal dominant families they have studied it would be useful to include this information for future genetic counseling protocols. They do comment in table 3 that one individual had their phenotype changed in 2020 on further examination so if that was mild and especially if post sequence results that is important for analyses that will be done on infants.

As you note, one of the probands was only found to be affected upon repeat clinical evaluation as an adolescent, which occurred after the SF3B2 mutation was known. To address this, we have added the following to the discussion:

‘While there were no clear examples of incomplete penetrance in our cohort, one proband (Kindred 8; III:1) evaluated in infancy did not demonstrate features of CFM, however repeat clinical evaluation in adolescence after identification of the *SF3B2* variant revealed mandibular hypoplasia. These findings imply that those with *SF3B2* LOF mutations identified in infancy with no obvious manifestations of the condition should be assessed closely throughout their development for delayed, potentially mild presentations of CFM.’

This pedigree also has both L and R sided affecteds which is relevant to the comment below on sidedness. Similarly are there any examples of chromosomal deletions (or duplications) of the SF3B2 locus that could further support the haploinsufficiency model?

Whereas we did not identify any structural variants encompassing SF3B2 in our cohort, analysis of public repositories identify a *de novo* CNV encompassing SF3B2 in a proband with CFM. The following has been added to the text to add this information:

We assessed variant calls from probands for rare, damaging (as called by Meta-SVM) missense variants in *SF3B2*, as well as structural variants encompassing the gene, however none were identified. Analysis of structural variants in the DECIPHER database identified a child with a 1Mb deletion encompassing *SF3B2* presenting with features of CFM (microtia, preauricular skin tag,

other craniofacial dysmorphism,[16, 17] suggesting that deletions encompassing the gene could also cause some cases of CFM.

Next in supplementary figure 2 one family with the IVS15-2 label has Sanger sequencing that appears to show that the mother has the same mutation as the proband but it is not boxed or commented on. I could also not align that table and its labels with table 1 in the main text. Please clarify what is going on here as in is that mother the mosaic they refer to? One of the dominant families? The labeling should be consistent.

We apologize for the error identified here; the proper Sanger traces for this family are now present, identifying a *de novo* single base deletion affecting a canonical splice site. We have relabeled each set of traces to identify the corresponding Kindred ID assigned in the text and tables, and placed these traces in numerical order.

See below for the xenopus work, but do they have any hypotheses why, even in the dominant families, this disorder is most commonly unilateral? This is one of its truly distinguishing features of CFM and is there data from mouse Kos or even mouse expression studies that show, for example, asymmetric expression or is SF3B2 in a pathway that includes other genes that have sidedness in their expression.

This is, perhaps, the “million dollar question” in CFM. We do not have any indication at this time as to why CFM is unilateral in so many cases, and our expression studies don’t identify asymmetric expression of *SF3B2*. We have added the following to the discussion though to highlight some potential theories: ‘Notably, the often unilateral nature of disease in CFM remains enigmatic. Other common craniofacial anomalies, including cleft lip and craniosynostosis, similarly present with unilateral disease more frequently than bilateral disease. Popular theories to explain this enigma include somatic “second hits” in genes wherein haploinsufficient germline variants predispose to disease, and genetic modifiers of phenotypic presentation. Large families demonstrating substantial phenotypic variation, much like the multiplex kindred presented above, provide evidence against somatic events causing the disease phenotype in the case of *SF3B2*-associated CFM, as the recurrence of such events in a single gene across multiple generations would be highly unlikely. The potential contribution of somatic mutations in other genes, as well as genetic modifiers of the phenotype, merit further investigation in CFM and other congenital craniofacial anomalies.’

In the discussion there was no note of how their prior work on how MYT1 fits into this picture both in terms of its clinical contributions (for example do any of the patients with SF3B2 variants also have MYT1 variants or is MYT1 in any pathways common with SF3B2? Since the overall numbers explained by SF3B2 (5%) are modest helping to focus the search next for future genetic contributions could be a useful discussion point.

Using the well-validated gene discovery technique and allele frequency thresholds described in the text ($MAF < 2 \times 10^{-5}$ in GnomAD, LOF or D-mis as called by MetaSVM), only a single rare, damaging (D-mis or LOF) variant in *MYT1* was identified in this cohort. The variant, a nonsense mutation, was identified in a multiplex kindred, but did not segregate with the affected individuals; thus, these specific data cannot currently support a role for *MYT1* variants in CFM. We do, however, provide some insight into potential further avenues to identify other genes or pathways that contribute to CFM, as suggested by yourself and another reviewer. We have added to our results a burden analysis demonstrating that when excluding kindreds with *SF3B2* variants, our discovery cohort has a persistently significant excess of *de novo* LOF mutations in cases (1.6-fold excess; $P=0.03$, Poisson test; Supplementary Table 1). We also note that given the observed excess

of *de novo* LOFs in the remaining 136 trios, we expect that an additional ~5.4% of cases might be explained by these mutations, with description of this calculation added to the Methods section and supporting data added to the supplement. We have added mention in our concluding paragraph that given the apparently high degree of locus heterogeneity in CFM, the results suggest that sequencing a substantially larger cohort of CFM cases is likely to reveal other genes or pathways in which *de novo* mutations contribute to disease risk.

They also present supportive data from an animal model (xenopus) for this finding. Here the story is perhaps less rigorous, so at least the authors might do some specificity controls for their antisense knockdown (see below, rescue and show expression of the gene in the neural crest or ubiquitous). Showing more mechanism in the xenopus models would improve this section. Although, they show the *sox10*-inhibition phenotype is very similar to that of *Sf3b4* knockdown (here and in a previous paper); and since the syndromes seems so similar and the proteins are proposed to act together in the U2 snRNP complex, it's unclear whether this represents a significant advancement. Also, unfortunately the frog work probably can't speak much to the unilaterality of the human phenotype since their effects are all from unilateral injections. Overall these assays are useful for working out mechanisms of the genes involved and for finding new candidates. However, the extent that *Sf3b2/4*-depletion of neural crest cell populations precisely mimics the human disease etiology (or whether these genes are also involved in different/more differentiated cell types later on) would require tissue-specific and/or temporal genetic ablation.

The specificity and efficacy of the antisense morpholino oligo (MO) was not tested for by the usual methods: 1) showing the MO can block in vitro translation of the RNA, 2) rescue of the phenotype with injected mRNA that is not bound by the MO (can be the human homologue) or using two non-overlapping MOs (not preferred—only if the injected RNA causes abnormalities). Also, the authors didn't show that the gene is even expressed in the tissue of interest (ie, neural crest). The labs involved here can do all those experiments and have done so in prior work on *SF3B4* (involved in the similar Nager and Rodriguez Syndromes, Ref. 20 in the submitted paper, Devotta et al., 2016), which could explain the apparently stronger effect of *Sf3b2* depletion. The authors may also consider testing putative loss-of-function variants (ie, point mutants or small indels) by showing failure to rescue the MO depletion (as in Devotta et al). Since both the *Sf3b2/4* genes are in the U2 snRNP complex, it would be nice to see some more parallel studies to see how similar the phenotypes are (*Sf3b4* depletion suggested increased apoptosis as a possible mechanism for neural crest depletion). The structural model seems to suggest that *b2* might link *b4* to the complex (Sun, C. The SF3b complex: splicing and beyond. *Cell. Mol. Life Sci.* 77, 3583–3595 (2020). <https://doi.org/10.1007/s00018-020-03493-z>), so the *b2* depletion might be stronger (maybe explaining the dominant inheritance?). Many labs also test putative loss-of-function variants by showing failure to rescue the MO depletion (also in Devotta et al).

We thank the reviewer for the thoughtful comments and suggestions. To address specificity of the morphant phenotype we have now performed rescue experiments by injection of human *SF3B2*. We show that increasing doses of *SF3B2* (10pg and 100pg) can restore *sox10* expression in morphant embryos (Fig 3A,B). In all our injections we check for toxicity and potential developmental delay by injecting a similar amount of a random control morpholino (CoMO). We have also included new data describing the expression of *sf3b2* in *Xenopus* embryos (Supplementary Figure 1). We appreciate the suggestion to perform rescue experiments with *Sf3b2* variants, and studies to compare the phenotype of *Sf3b2* and *Sf3b4*, however we feel that this is beyond the scope of the work.

Other xenopus issues include:

- The protein domain diagram could be modified to include proline-rich and glutamine-rich regions in the

N- and C-terminal regions, respectively (and other relevant information from the NCBI protein entry).

The protein domain included in the manuscript includes the domains listed in the human Uniprot entry consistent across submissions, as the mutations identified represent those identified in human subjects. While the proline- and glutamine-rich regions aren't annotated in the human submissions, we would be happy to include mention of any domains in the human gene that this reviewer feels might be particularly relevant.

- In two instances on p.5 the authors refer to proliferation defects in depleted embryos; studies on proliferation are not presented and thus these statements are not supported by the data (also, the similar phenotypes of Sf3b4-depletions were linked to increased apoptosis, not proliferation, in a previous paper (Devotta, et al., 2016).

We agree, it is an overstatement and we have revised this sentence to more accurately reflect our findings, we are now referring to “defects in neural crest precursors formation”.

- Similarly, the section on ‘poison exons’ seems speculative, especially since other data from the same paper on Sf3b4 did not show indications of splicing defects, at least on selected neural crest genes.

As you note, this is entirely speculative, and was intended to be presented as a mere possibility within the discussion section as we believe that it merits further investigation; we state “it will be interesting to determine whether retention of poison exons via aberrant splicing might contribute to this disease process.” We hope to employ further studies to analyze splicing in tissues from affected patients, which would include assessing for retention of poison exons. If this reviewer feels that our statement is misleading, we would be happy to change the wording further.

Minor issues:

The phrase “linkage of the trait to chromosomes 11q12-13 and 14q32 ” might reframe chromosome to chromosomal bands.

Thank you for this recommendation- we have changed the text to note that these are chromosomal bands.

Missing word, line 5 Introductory paragraph: “. . .nuclear ribonucleoprotein ^, in probands. Twenty . . .”;
^ complex

Thank you for identifying this word omission, which has been fixed in the introduction.

Xenopus gene names and symbols are lower case and italics by community convention (<http://www.xenbase.org/entry/>); sf3b2

Thank you for noting this. We have incorporated proper *Xenopus* gene notation when describing the animal studies.

Overall this is a compelling report that mutations, both familial and de novo, can cause CFM in about 5% of cases, that the SF3B2 gene involved is a logical candidate given its role in neural crest, its paralogs role in Nager S and its interactions with other relevant developmental pathways. It has been carried out carefully, with a very large sample size for a rare disorder and is a quantum advance in our biological understanding of CFM and has important immediate clinical implications for genetic counseling and opens new doors to future biological understanding.

We thank this reviewer for the time and effort put into this thoughtful review. Our manuscript has

been significantly improved by incorporating the recommended studies and revisions.

Reviewer #2 (Remarks to the Author):

The authors have made substantial revisions to their manuscript to address primary genetic concerns related to mutations, cohorts, kindreds, frequencies, prevalence etc which have improved the manuscript. I appreciated the inclusion of new analyses of additional neural crest cell markers and demonstration of a cartilage phenotype following MO knockdown which goes a long way to connect SF3B2 to the progenitor cell population that generates the tissues that are effected in CFM.

Although it would have been preferable to have more mechanism, I can respect the authors opinion that it's beyond the scope of the current paper. But it does mean the authors need to modify their abstract. Currently they state in the abstract that their data supports a link between spliceosome dysfunction and impaired neural crest cell development. It's becoming increasingly clear for many ubiquitously expressed factors that regulate global functions, that many have functions outside their traditional roles. Therefore without analysis of spliceosome function or splicing data this sort of definitive statement about spliceosomal dysfunction should not be in the abstract. In contrast, it's fine to include it in the discussion as a speculative interpretation.

Reviewer #3 (Remarks to the Author):

The authors have responded well to the critiques of both this specific reviewer and apparently to those of the other reviewers as well. Overall they have corrected a few errors, modified language (and percentages) to better reflect the data and enhanced the discussion to be more effective and to foreshadow future studies by themselves and others. For the human data the only concern would be tying their findings of rare variants in CFM to work on cleft lip and palate. The added discussion around IRF6 and two references is not quite analogous to the findings here in that the sequencing of non-syndromic clefts may have included true autosomal dominant Van Der Woude families that merely lacked the defining phenotypic feature of lip pits (as happens in ~10% of cases) and thus those cases were just phenotypic variants where the affected would have a 50% chance of having an affected child. Nonetheless IRF6 does show up in GWAS studies of non syndromic clefting quite strongly. The authors here would have an opportunity in the future to evaluate whether common variants in SF3B2 are over transmitted via a TDT test on their isolated families (and the TDT can be quite powerful even when the number of families is modest by GWAS standards) as a candidate gene and indeed they could even do true GWAS on CFM here as again although the number of families is small using the TDT they should still have some power to detect a genome wide signal of a non SF3B2 gene.

For the Xenopus work the revised manuscript has also adequately addressed the concerns regarding the Xenopus experiments. There are only a few minor/clarifying comments: (1) please check the congruency between Fig. 3 and its legend, particularly for panel A — the legend mentions increasing doses and comparison with “SF3B4 knockdown” but the middle of panel shows two sets of SF3B2 MO embryos. Are these different doses of B2 MO or a comparison with B4 MO? Also, I'm guessing the two rows are just different examples of the effect on sox10, but this could be pointed out for the benefit of the non-Xenopus-oriented readers (same with Fig. 4A), and (2) the authors may want to clarify how the ANOVA analysis was done; was this just for the high dose of SF3B2 MO or all the experiments? Post hoc tests?

Overall the manuscript is substantially improved and has addressed the concerns of these reviewers

and will provide an important advance in the field.

Doug Houston

Jeff Murray

RESPONSE TO REVIEWERS

REVIEWERS' COMMENTS

Reviewer #2 (Remarks to the Author):

The authors have made substantial revisions to their manuscript to address primary genetic concerns related to mutations, cohorts, kindreds, frequencies, prevalence etc which have improved the manuscript. I appreciated the inclusion of new analyses of additional neural crest cell markers and demonstration of a cartilage phenotype following MO knockdown which goes a long way to connect SF3B2 to the progenitor cell population that generates the tissues that are effected in CFM.

We thank this reviewer for their comments.

Although it would have been preferable to have more mechanism, I can respect the authors opinion that it's beyond the scope of the current paper. But it does mean the authors need to modify their abstract. Currently they state in the abstract that their data supports a link between spliceosome dysfunction and impaired neural crest cell development. It's becoming increasingly clear for many ubiquitously expressed factors that regulate global functions, that many have functions outside their traditional roles. Therefore without analysis of spliceosome function or splicing data this sort of definitive statement about spliceosomal dysfunction should not be in the abstract. In contrast, it's fine to include it in the discussion as a speculative interpretation.

We appreciate this comment and have changed the abstract accordingly; instead of referring to spliceosomal dysfunction, we now refer to spliceosome mutations, as we have identified mutations in SF3B2 as causal in this disease.

Reviewer #3 (Remarks to the Author):

The authors have responded well to the critiques of both this specific reviewer and apparently to those of the other reviewers as well. Overall they have corrected a few errors, modified language (and percentages) to better reflect the data and enhanced the discussion to be more effective and to foreshadow future studies by themselves and others. For the human data the only concern would be tying their findings of rare variants in CFM to work on cleft lip and palate. The added discussion around IRF6 and two references is not quite analogous to the findings here in that the sequencing of non-syndromic clefts may have included true autosomal dominant Van Der Woude families that merely lacked the defining phenotypic feature of lip pits (as happens in ~10% of cases) and thus those cases were just phenotypic variants where the affected would have a 50% chance of having an affected child. Nonetheless IRF6 does show up in GWAS studies of non syndromic clefting quite strongly. The authors here would have an opportunity in the future to evaluate whether common variants in SF3B2 are over transmitted via a TDT test on their isolated families (and the TDT can be quite powerful even when the number of families is modest by GWAS standards) as a candidate gene and indeed they could even do true GWAS on CFM here as again although the number of families is small using the TDT they should still have some power to detect a genome wide signal of a non SF3B2 gene.

We thank these authors for their comments. We must disagree regarding the relation to IRF6, as we believe it is actually quite similar. Much like in the “non-syndromic” IRF6 probands, these probands with SF3B2 mutations do not have any features that indicate that they have a distinct form of craniofacial microsomia, however our results imply that they as well would have a 50% chance of having an affected child as they too demonstrate autosomal dominant inheritance. Thus,

dominant genes (IRF6, SMAD6, SF3B2) appear to play a role in ~5% of “non-syndromic” cases of the craniofacial malformations described (cleft lip, craniosynostosis, craniofacial microsomia), each with important implications for genetic counseling as we describe in the discussion. Similar to IRF6 probands in cleft lip, SF3B2 probands with CFM should be counseled regarding the 50% likelihood of having an affected child, albeit with variable penetrance. While performing a larger scale GWAS is an avenue of interest to us in the future, we are underpowered to do so at this time.

For the *Xenopus* work the revised manuscript has also adequately addressed the concerns regarding the *Xenopus* experiments. There are only a few minor/clarifying comments: (1) please check the congruency between Fig. 3 and its legend, particularly for panel A — the legend mentions increasing doses and comparison with “SF3B4 knockdown” but the middle of panel shows two sets of SF3B2 MO embryos. Are these different doses of B2 MO or a comparison with B4 MO? Also, I’m guessing the two rows are just different examples of the effect on *sox10*, but this could be pointed out for the benefit of the non-*Xenopus*-oriented readers (same with Fig. 4A), and (2) the authors may want to clarify how the ANOVA analysis was done; was this just for the high dose of SF3B2 MO or all the experiments? Post hoc tests?

Thank you for pointing that out, Figure 3 does require clarification. In panel A, the second column is showing Sf3b4MO-injected embryos for comparison. This has been corrected. The reviewer is correct, the two rows in Figure 3A and Figure 4A are showing examples of the effect of the MO injection. This has been clarified in the figure legend for both Figures. Regarding the ANOVA, all experiments were included on a one-way ANOVA. This has been clarified in the text. No post hoc tests were performed.

Overall the manuscript is substantially improved and has addressed the concerns of these reviewers and will provide an important advance in the field.

Doug Houston
Jeff Murray